# Integrated NMR and MS Analysis of the Plasma Metabolome Reveals Major Changes in One-Carbon, Lipid, and Amino Acid Metabolism in Severe and Fatal Cases of COVID-19

**DOI:** 10.3390/metabo13070879

**Published:** 2023-07-24

**Authors:** Marcos C. Gama-Almeida, Gabriela D. A. Pinto, Lívia Teixeira, Eugenio D. Hottz, Paula Ivens, Hygor Ribeiro, Rafael Garrett, Alexandre G. Torres, Talita I. A. Carneiro, Bianca de O. Barbalho, Christian Ludwig, Claudio J. Struchiner, Iranaia Assunção-Miranda, Ana Paula C. Valente, Fernando A. Bozza, Patrícia T. Bozza, Gilson C. dos Santos, Tatiana El-Bacha

**Affiliations:** 1LeBioME-Bioactives, Mitochondrial and Placental Metabolism Core, Institute of Nutrition Josué de Castro, Universidade Federal do Rio de Janeiro, Rio de Janeiro 21941-902, Brazil; marcosalmeidajj@gmail.com (M.C.G.-A.); gabidap@gmail.com (G.D.A.P.); torres@iq.ufrj.br (A.G.T.); talita.tiac@gmail.com (T.I.A.C.); biancabarbalho@ufrj.br (B.d.O.B.); 2Laboratory of Immunopharmacology, Oswaldo Cruz Institute, Oswaldo Cruz Foundation, Rio de Janeiro 21041-361, Brazil; liviaimunofar@gmail.com (L.T.); pbozza@ioc.fiocruz.br (P.T.B.); 3Laboratory of Immunothrombosis, Department of Biochemistry, Federal University of Juiz de Fora, Juiz de Fora 36936-900, Brazil; eugenio.hottz@icb.ufjf.br; 4LabMeta, Metabolomics Laboratory, Institute of Chemistry, Universidade Federal do Rio de Janeiro, Rio de Janeiro 21941-598, Brazil; ivenspaula@gmail.com (P.I.); ribeirohygor@ufrj.br (H.R.); rafael_garrett@iq.ufrj.br (R.G.); 5Lipid Biochemistry and Lipidomics Laboratory, Department of Chemistry, Universidade Federal do Rio de Janeiro, Rio de Janeiro 21941-598, Brazil; 6Institute of Metabolism and Systems Research, University of Birmingham, Birmingham B15 2SQ, UK; c.ludwig@bham.ac.uk; 7School of Applied Mathematics, Fundação Getúlio Vargas, Rio de Janeiro 22231-080, Brazil; claudio.struchiner@fgv.br; 8Institute of Social Medicine, Universidade do Estado do Rio de Janeiro, Rio de Janeiro 20550-013, Brazil; 9LaRIV, Instituto de Microbiologia Paulo de Goes, Universidade Federal do Rio de Janeiro, Rio de Janeiro 21941-902, Brazil; iranaiamiranda@micro.ufrj.br; 10National Center for Nuclear Magnetic Resonance—Jiri Jonas, Institute of Medical Biochemistry, Universidade Federal do Rio de Janeiro, Rio de Janeiro 21941-902, Brazil; valente@cnrmn.bioqmed.ufrj.br; 11National Institute of Infectious Disease Evandro Chagas, Oswaldo Cruz Foundation, Rio de Janeiro 21040-360, Brazil; fernando.bozza@ini.fiocruz.br; 12D’Or Institute for Research and Education, Rio de Janeiro 22281-100, Brazil; 13LabMet-Laboratory of Metabolomics, Instituto de Biologia Roberto Alcantara Gomes (IBRAG), Department of Genetics, State University of Rio de Janeiro, Rio de Janeiro 20551-030, Brazil

**Keywords:** SARS-CoV-2, metabolomics, ^1^H-NMR, high-resolution mass spectrometry, fatal COVID-19, virus-host interactions, metabolic alterations, sex differences

## Abstract

Brazil has the second-highest COVID-19 death rate worldwide, and Rio de Janeiro is among the states with the highest rate in the country. Although vaccine coverage has been achieved, it is anticipated that COVID-19 will transition into an endemic disease. It is concerning that the molecular mechanisms underlying clinical evolution from mild to severe disease, as well as the mechanisms leading to long COVID-19, are not yet fully understood. NMR and MS-based metabolomics were used to identify metabolites associated with COVID-19 pathophysiology and disease outcome. Severe COVID-19 cases (n = 35) were enrolled in two reference centers in Rio de Janeiro within 72 h of ICU admission, alongside 12 non-infected control subjects. COVID-19 patients were grouped into survivors (n = 18) and non-survivors (n = 17). Choline-related metabolites, serine, glycine, and betaine, were reduced in severe COVID-19, indicating dysregulation in methyl donors. Non-survivors had higher levels of creatine/creatinine, 4-hydroxyproline, gluconic acid, and *N*-acetylserine, indicating liver and kidney dysfunction. Several changes were greater in women; thus, patients’ sex should be considered in pandemic surveillance to achieve better disease stratification and improve outcomes. These metabolic alterations may be useful to monitor organ (dys) function and to understand the pathophysiology of acute and possibly post-acute COVID-19 syndromes.

## 1. Introduction

The existing vaccines for SARS-CoV-2 infection resulted in a significant reduction in the number of severe cases of the disease. However, it is anticipated that the coronavirus disease 2019 (COVID-19) will transition into an endemic state [1]. By the time this article is being written, COVID-19 will have exceeded 767 million cases with more than 6.9 million deaths worldwide [2]. Brazil has recorded more than 700,000 deaths, making it the country with the second-highest number of deaths worldwide [2]. Rio de Janeiro, where this study took place, had 2.8 million confirmed cases with more than 77,000 deaths. The state is among the ones with the highest mortality rate [3].

The replication of SARS-CoV-2 triggers a systemic immune response that leads to tissue damage and the reprogramming of whole-body metabolism [4,5]. Additionally, around 20% of infected subjects may experience long-term symptoms after recovery from the initial illness [6], a condition that is associated with neurological, gastrointestinal, pulmonary, and cardiovascular alterations and which can be highly debilitating [7]. Even patients who present mild symptoms in the acute phase of the disease may later develop post-acute COVID-19 syndrome [8]. 

These observations reveal the complexity of COVID-19 pathophysiology and its profound impact on different tissues and cells. However, the underlying molecular mechanisms leading to clinical evolution from mild to severe disease, as well as the mechanisms associated with post-acute COVID-19 symptoms, are not yet known. The complex multisystemic nature of SARS-CoV-2 infection calls for a system-level approach that provides a better understanding of the molecular mechanisms underlying COVID-19 pathophysiology.

Metabolomics is the central omics in information translation [9], providing the metabolic signature of organs and biological fluids in different conditions. We and others have shown that metabolomics was essential in the development of new approaches that improved the understanding, therapeutics, and clinical management of emerging viral diseases such as Dengue [10,11], Chikungunya [10], SARS [12], and Zika [13,14], and may be helpful in elucidating the metabolic pathways associated with COVID-19. Alterations in host energy, amino acid, and lipid metabolism are frequently observed in viral infections as the virus disturbs and exploits host metabolic pathways for its own benefit [15,16]. These metabolic alterations have a critical role in disease outcome and in modulating the host immune response. 

The modulation of host lipid metabolism is a feature shared by coronaviruses and is essential for viral RNA replication [17], as it enables the synthesis of the viral envelope membrane as well as double-membrane vesicles and lipid compartments. Indeed, our group has shown that lipid droplets accumulate in monocytes isolated from COVID-19 subjects and serve as an assembly platform for SARS-CoV-2 particles [18]. The orchestration of lipid flow within different cell compartments by the SARS-CoV-2 non-structural protein 6 ensures the proper organization of double-membrane vesicles as well as their effective communication with lipid droplets [19]. All these events are essential for SARS-CoV-2 replication. 

One of the first studies to use a multi-omics approach to gain insight into the pathophysiology of COVID-19 was performed by Shen et al. with a small cohort of patients. In that study, proteomics and metabolomics approaches revealed that mild and severe COVID-19 patients presented metabolic and immune dysregulation [20]. More than 100 lipid species, including glycerophospholipids, fatty acids, lipoproteins, and several amino acids, were downregulated in sera from subjects infected with SARS-CoV-2 if compared to controls [20]. Alterations in lipoproteins using NMR metabolomics were also confirmed in larger cohorts [21], and single-cell metabolomics of monocytes reinforced the idea that modulation of intermediary metabolism, in particular organic acids, plays crucial roles in COVID-19 severity [22]. Regarding the modulation of amino acid metabolism, several studies have reported that patients have low levels of plasma tryptophan, which is now considered a marker for the extent of inflammation and COVID-19 severity [20,23,24,25]. Additionally, it has been reported that COVID-19 inpatients show dysregulation in the metabolism of methyl donors, including higher levels of S-adenosyl-homocysteine and lower levels of homocysteine, regardless of IL-6 levels [26]. Indeed, SARS-CoV-2 genome replication seems to depend on folate and methionine cycle modulation [27]. Therefore, SARS-CoV-2 infection is thought to affect various aspects of host metabolism, and the extent of these changes is believed to be linked to the severity of the disease. On the other hand, the specific metabolic differences that may distinguish the severe cases of COVID-19 from those that are fatal have not yet been fully addressed. 

The purpose of this study was to characterize in depth the metabolic alterations associated with the severe cases of COVID-19 and to investigate potential markers of fatal outcome. ^1^H Nuclear Magnetic Resonance (NMR) spectroscopy and Liquid Chromatography-High-Resolution Mass spectrometry (LC-HRMS)-based metabolomics were used in a well characterized prospective cohort of subjects with severe COVID-19, including survivors and non-survivors, and healthy subjects. Patients’ samples were collected in Rio de Janeiro, Brazil, between April and July 2020. We were particularly interested in investigating metabolites associated with one-carbon metabolism and with lipid and amino acid metabolism. Considering that the incidence of post-acute severe outcomes after hospital discharge is very high after severe COVID-19, also in Brazil [28], these metabolic alterations may be useful to monitor patients’ organs and tissues (dys)function and to understand acute pathophysiological mechanisms that may lead to post-acute COVID-19 syndrome [29,30,31,32]. 

## 2. Materials and Methods

### 2.1. Study Design and Participants 

We prospectively enrolled a cohort of 35 RT-PCR-confirmed severe COVID-19 cases within 72 h of intensive care unit (ICU) admission in two reference centers in Rio de Janeiro, Brazil (Instituto Estadual do Cérebro Paulo Niemeyer and Hospital Copa Star), between April and July 2020. Enrichment-dependent SARS-CoV-2 sequencing of a subsample of this cohort showed that over 70% of SARS-CoV-2 samples were phylogenetically related to the emerging clade 20B. Clades 19A and 20A were also detected [33].

All patients were adults (≥18 years of age) classified as having severe COVID-19 (n = 35) according to the WHO working group on the clinical characterization and management of COVID-19 [34]. Severe COVID-19 was defined as critically ill patients presenting with viral pneumonia confirmed by the presence of chest infiltrates on a computed tomography scan and by the need for respiratory support with either non-invasive oxygen supplementation or mechanical ventilation. The complete clinical information was collected prospectively using a standardized form: International Severe Acute Respiratory and Emerging Infection Consortium (ISARIC)/World Health Organization (WHO) Clinical Characterization Protocol for Severe Emerging Infections (CCP-BR). Upon admission, clinical and laboratory data were recorded for all severe patients included in the study. The primary outcome analyzed was 28-day mortality, and patients were classified as survivors (n = 18) or non-survivors (n = 17).

All ICU-admitted patients received the usual supportive care for severe COVID-19. Patients with acute respiratory distress syndrome (ARDS) were managed with neuromuscular blockade and a protective ventilation strategy that included low tidal volume (6 mL/kg predicted body weight) and limited driving pressure (<16 cm H_2_O) as well as optimal positive end-expiratory pressure calculated based on the best lung compliance and PaO_2_/fraction of inspired oxygen (FiO_2_) ratio. A prone position was adopted in those patients with severe ARDS and a PaO_2_/FiO_2_ ratio < 150. Antithrombotic prophylaxis was performed with 40 to 60 mg of enoxaparin per day. Patients did not receive antivirals, steroids, or other anti-inflammatory or antiplatelet drugs in accordance with clinical practice at the time of inclusion.

Peripheral blood was also collected from SARS-CoV-2-negative participants (control group; n = 12), confirmed by RT-PCR of nasal swabs on the day of blood sampling. The control group included subjects of matching age and sex distribution compared to infected subjects. These participants had not been on anti-inflammatory or antiplatelet drugs for at least 2 weeks prior to the study. 

The study was conducted according to the guidelines of the Declaration of Helsinki and approved by the National Review Board of Brazil (Comissão Nacional de Ética em Pesquisa [CONEP] 30650420.4.1001.0008), and informed consent was obtained from all subjects or their caregivers.

### 2.2. Chemicals and Solvents

All solvents used were of HPLC analytical grade. Acetonitrile and methanol were obtained from TEDIA^®^ (Fairfield, OH, USA), and isopropanol from Sigma Aldrich (São Paulo, Brazil). Water was purified in the Milli-Q device, the Millipore Purification System (Billerica, MA, USA). Mobile phase additives formic acid and ammonium hydroxide were purchased from TEDIA^®^, and ammonium acetate was obtained from J. T. Baker^®^ (Aparecida de Goiânia, Brazil). Isotopically labeled internal standards, U ^13^C D-glucose and U ^13^C L-glutamine, and deuterium oxide were purchased from Cambridge Isotope Laboratories, Inc. (Tewksbury, MA, USA). All other standards were obtained from Sigma Aldrich. 

### 2.3. Sample Processing

Blood samples were drawn into acid-citrate-dextrose and centrifuged (200× *g*, 20 min, room temperature). Plasma was collected and stored at −80 °C until analysis. A citrate-dextrose buffer was chosen to preserve platelets. This choice of buffer prevented us from comparing citrate and sugars among groups and limited the identification of metabolites that present chemical shifts in the ^1^H NMR spectrum, which is in the vicinity of citrate. 

### 2.4. Nuclear Magnetic Resonance-Based Metabolomics

#### 2.4.1. Sample Preparation

Frozen plasma samples were quickly thawed and diluted 3-fold in sodium phosphate buffer and deuterium oxide (final concentration: 50 mM phosphate buffer and 10% deuterium oxide, pH 7.4). A total of 600 μL of diluted samples were transferred to a 5 mm NMR tube.

#### 2.4.2. NMR Acquisition, Spectra Pre-Processing, and Metabolite Assignment

NMR spectra were acquired on a Bruker Advance III at 500.13 MHz at 300 K, coupled with a cooled automatic sample case at 280 K. 1D-^1^H NMR spectra were acquired using excitation sculpting to suppress the solvent signal [35] as well as a CPMG (Carr-Purcell-Meiboom-Gill) T2 filter [36] with 32 loop counters and a delay of 0.001 s. 32,768 complex data points were acquired per transient, for a total of 1024 transients. The spectral width was set to 19.99 ppm, resulting in an acquisition time of 3.27 s per FID. The relaxation delay was set to 1.74 s. 

Spectra data were pre-processed in the MetaboLab [37] software v. 2022.0726.1733. Prior to the Fourier transform, the FIDs were apodized using an exponential window function with 0.3 Hz line-broadening and then zero-filled to 65,536 data points. After Fourier transform, each spectrum was manually phase corrected, followed by a spline-baseline correction. Finally, all spectra were referenced to the signal of the ^1^H linked to the anomeric carbon of glucose. Baseline noise and regions corresponding to water and citrate signals were deleted. Spectra data were binned with a 0.005 ppm interval, and the resulting table presented 81,498 data points, corresponding to metabolites’ intensities. A generalized log transformation [38] was applied prior to multivariate statistical analysis.

Following 2D spectra, HSQC ^1^H-^13^C and TOCSY ^1^H-^1^H acquisition, data was uploaded on the COLMAR [39,40] for the assignments. The peak report of all assigned compounds can be seen at https://spin.ccic.osu.edu/index.php/colmarm, session ID 3121-pZ5ZukwXBh, (accessed on 14 July 2023) (COLMAR https://spin.ccic.osu.edu/index.php/colmarm/index2, accessed on 14 July 2023). Appendix A present the ^1^H NMR assignment information for the metabolites and broad signals of lipids and proteins that distinguished the groups.

We also overlaid spectra from the BMRB [41] and HMDB 4.0 [42] databanks. The software ICON NMR (Bruker) was used for automatic acquisition.

### 2.5. Mass Spectrometry-Based Metabolomics

#### 2.5.1. Standards

A stock of internal standard (IS) solution was prepared with a final concentration of 0.15 mg mL^−1^ for U-^13^C D-glucose and 0.13 mg mL^−1^ for U-^13^C L-glutamine in acetonitrile/isopropanol/water (3:3:2, % *v*/*v*/*v*). 

Stock solutions of targeted analytes were prepared at 1.0 mg mL^−1^ in methanol or in different proportions of acetonitrile/water. A standard working solution was prepared by mixing appropriate volumes of each stock solution to reach the final concentration of 2.0–50.0 µg mL^−1^ in acetonitrile/water (1:1, %/v). 

#### 2.5.2. Sample Preparation

A total of 30 μL of plasma (in duplicate) were mixed with the same volume of the IS mixture and 500 μL of a degassed mixture of pre-chilled acetonitrile/isopropanol/water (3:3:2, *v*/*v*/*v*). After vortexing for 20 s and incubating in ice in an ultrasonic water bath for 5 min, samples were centrifuged at 12,000× *g* at 4 °C for 5 min, and the supernatant (480 μL) was dried under Nitrogen gas. Samples were reconstituted with 60 μL of acetonitrile/water (1:1, *v*/*v*) containing 2 μg mL^−1^ of the IS p-fluoro-L-phenylalanine, vortexed for 15 s, and centrifuged as above. The resulting supernatant was used for LC-MS analysis. 

A pooled quality control (QC) sample was prepared by combining 5 μL of each plasma before extraction and processing it in the same way as the specimen samples. QC samples were injected with every tenth biological sample to monitor the stability of the analytical system as well as the reproducibility of the procedure for sample treatment [43]. 

Subgroups of pooled QC samples (control, survivors, or non-survivors) were created to collect fragmentation spectra via data-dependent acquisition (DDA) mode on the mass spectrometer. The analysts running MS-based experiments were blinded to the sample grouping until the end of data analysis to limit biased peak annotation. 

### 2.6. LC-MS Conditions

Liquid chromatography (LC) analysis was performed on a Dionex UltiMate 3000 UHPLC (Thermo Fisher Scientific, Bremen, Germany) system using a Waters^®^ ACQUITY UPLC^®^ BEH amide column (150 × 2.1 mm × 1.7 μm) by gradient elution at a constant flow rate of 350 μL min^−1^. The column oven temperature and injection volume were set to 40 °C and 5.0 μL, respectively. Two different mobile phase compositions with different pH values were used. One consisted of (A) water:acetonitrile (95:5, *v*/*v*) and (B) acetonitrile:water (95:5, *v*/*v*) both with 0.1% formic acid (pH 3), and the other consisted of (A) water:acetonitrile (95:5, *v*/*v*) and (B) acetonitrile:water (95:5, *v*/*v*) both with 0.05% ammonium hydroxide and 10 mM ammonium acetate (pH 8). The gradient elution was 0–0.5 min 100% B; 0.5–5.0 min 45% B; 5.0–9.0 min 45% B; 9.0–10.0 min 100% B; 10.0–15.0 min 100% B. 

The LC was coupled to a hybrid Quadrupole-Orbitrap high-resolution and accurate mass spectrometer (QExactive Plus, Thermo Scientific, Waltham, MA, USA) equipped with a heated electrospray ion source operating in both negative (ESI−) and positive (ESI+) ionization modes. Source ionization parameters were: spray voltage −3.6 kV/+3.9 kV; capillary temperature 270 °C; probe heater temperature 380 °C; S-Lens RF level 50; sheath and auxiliary gases 50 and 10 (arbitrary units), respectively. Samples were analyzed in Full MS mode in the scan range of *m*/*z* 50–710 at a resolution of 70,000 FWHM (full width at half maximum). The automatic gain control (AGC) target was set at 1 × 10^−6^ with a maximum injection time (IT) of 150 ms. 

The solution of target analytes and the subgroup pooled QC samples were analyzed in Full MS followed by data-dependent acquisition (dd-MS2 top5 experiment) in the same scan range as above. For the full MS scan, the mass resolution was set to 17,500 FWHM with the following settings: AGC target of 1 × 10^−6^ and maximum IT of 80 ms. For the dd-MS2 scan, the mass resolution was set to 17,500 FWHM with the following settings: AGC target at 1 × 10^−5^, maximum IT of 50 ms, isolation window at *m*/*z* 1.2, normalized collision energy (NCE) of 15, 35 (ESI+), and 10, 30 (ESI−), intensity threshold at 1 × 10^−6^, exclude isotopes “on”, and dynamic exclusion of 10.0 s. 

### 2.7. Non-Targeted and Targeted LC-HRMS-Based Metabolomics

For the non-targeted analysis, the LC-HRMS data files were submitted to a metabolomics workflow using MS-DIAL software (RIKEN, version 4.80) [44] for data processing, including peak matching against an MS/MS library. The parameters used for both pH 8 and pH 3 analyses are described in Appendix A. Features were selected assuming coefficient variation (CV) % values less than 30% in QC samples and a Gaussian-like peak shape according to the protocols for quality control used in untargeted metabolomics [43,45]. Prior to multivariate statistics, MS data were normalized by the Total Ion Chromatogram and scaled using Pareto.

Compound annotation was carried out: (i) based on the MS/MS fragment comparison with the standard compounds; (ii) by comparing the aligned *m*/*z* ions with a mass error below 6 ppm to those available at the HMDB [42] and METLIN website; and (iii) by comparing the investigated MS/MS spectra with a similarity score ≥ 80% to those in the NIST 20 Tandem Mass Spectral Library and MassBank of North America using a customized MSP file in MS-DIAL. Furthermore, molecular formulas were determined using MS-FINDER (RIKEN, version 3.52) [46]. 

Targeted data analysis was performed as a confirmation step for the non-targeted approach in TraceFinder software v3.1 (ThermoFisher Scientific, Waltham, MA, USA). An in-house library that includes retention time, exact mass, and fragments of the target compounds was used. As identification criteria, mass errors less than 5 ppm and retention time variations of <1% compared to the defined retention time were accepted [47]. Appendix A present the target compounds monitored via the method at pH 8 and pH 3, respectively.

### 2.8. Lipoprotein Analysis

Total cholesterol (TC), high-density lipoprotein (HDL), and triglycerides (TGs) were measured by the oxidase-peroxidase method [48,49]. Low-density lipoprotein (LDL) was calculated based on Friedewald’s equation (LDL-c = TC − HDL-c − TG/5).

### 2.9. Statistical Analysis

The sample size was determined by the feasibility of recruitment and eligibility criteria. 

Data distribution was analyzed using the Shapiro-Wilk test and median and interquartile intervals, and the Mann-Whitney or Kruskal-Wallis tests were used for data with an asymmetrical distribution. Categorical variables were compared using the Fisher’s exact test with absolute (n) and relative (%) frequencies. 

Data derived from processed NMR spectra was subjected to multivariate Principal Component Analysis using the MetaboAnalyst 4.0 [50] platform. For the univariate statistics, Kruskal-Wallis and Dunn’s post-hoc tests were used for variables’ comparison of non-transformed NMR or MS data. The interaction between sex and disease was analyzed using a two-way ANOVA and Tukey’s multiple comparison tests. GraphPad Prism version 8.4.3 was used for all analyses. *p* < 0.05 was considered for rejection of the null hypothesis. 

Classification and regression tree (CART) models [51] were fitted to assess which metabolites best predicted the observed morbidity class of study participants. The model-fitting algorithms have been implemented in the library “rpart” [52] for the R programming language. We set “method” = “class”, and the remaining parameters were kept at their default values for all models. CART models were built using the significant variables identified in the NMR and MS-based metabolomics (considering separate and combined datasets), in addition to the subjects’ sex and age. For the combination of NMR and MS results, a unified matrix was built by normalizing the data as their z-score.

## 3. Results

### 3.1. Subjects’ Demographics and Clinical Parameters

A total of 47 individuals were included in this study: 12 non-infected control subjects and 35 severe COVID-19 cases, grouped into survivors (n = 18) and non-survivors (n = 17) according to the 28-day mortality outcome. The demographics and clinical characteristics of all subjects included in the study are shown in Table 1. Briefly, age, sex distribution, and co-morbidities were similar among groups. Non-invasive oxygen supplementation was used in 44% of subjects in the survivors group, whereas 100% of the subjects in the non-survivors group received mechanical ventilation. 

Laboratory findings revealed that patients in the survivors group presented ~50% more monocyte counts (*p* = 0.009) if compared to non-survivors. At admission to the ICU, leukocyte and platelet counts were similar between the two groups of infected patients. 

### 3.2. ^1^H NMR- and MS-Based Metabolomics

Representative spectra of aliphatic (Figure 1A,B), amidic, and aromatic (Figure 1C) regions indicate important differences in the metabolite profile among the three groups. Discriminating metabolites such as (CH_3_)_3_ choline-related metabolites, creatine/creatinine, amino acids, organic acids, and broad residual signals of lipids are depicted, followed by arrows indicating higher/lower contents in severe COVID-19 cases. A discriminating metabolite profile was confirmed with the Principal Component Analysis (PCA) scores plot (Figure 1D), where principal components 1, 2, and 3 accounted for approximately 70% of the variation among groups. PCA loading factors plot highlights (CH_3_)_3_ choline, creatine/creatinine, lactate, acetate, and broad signals of CH_3_ and CH_2_ lipoproteins as discriminating variables (Appendix A).

To gain meaningful insights into the changes associated with disease severity and outcome, metabolites that exhibited significant differences in content among groups according to the PCA results were selected. NMR-based metabolomics revealed that the levels of (CH_3_)_3_-choline metabolites, including glycerophosphocholine, phosphocholine, and choline, as well as glycine, which is associated with one-carbon metabolism, were significantly lower in both survivors and non-survivors if compared to controls (Figure 2A,B). Additionally, ^1^H NMR-based metabolomics identified several metabolites related to glucose, insulin sensitivity, and inflammation that were significantly higher in infected subjects (Figure 2C–G), including creatine/creatinine (considered together due to signal overlap), *N*-acetyl ^1^H of glycoproteins, and lactate. Importantly, creatine/creatinine levels at admission set non-survivors apart from survivors and control subjects, as the levels were higher in patients that expired after up to 28 days of ICU stay. Lower levels of acetate and higher levels of formate, a byproduct of bacterial metabolism in the gut, were observed in infected subjects but not in controls (Figure 2F,G). Additionally, residual signals of (CH_2_) of VLDL-lipoproteins, lipids (CH=CH olefinic protons of triacylglycerols), and acetoacetate, a ketone body, were significantly higher in both groups of infected subjects if compared to controls (Figure 2H–J).

^1^H NMR-metabolomics findings also suggest that the dysregulation in amino acid metabolism is a function of severe COVID-19, as survivors and non-survivors had plasma levels of glutamine, alanine, branched-chain amino acids (valine, leucine, and isoleucine), and tyrosine lower than those observed in controls (Figure 2K–P). Appendix A present the ^1^H NMR assignment information for the metabolites and broad signals of lipids and proteins that distinguished the groups in both PCA and univariate analysis.

To further investigate the changes in the plasma metabolome associated with severe COVID-19, LC-high-resolution mass spectrometry (LC-HRMS)-based metabolomics was used. In the current study, LC-HRMS allowed us to confirm the alterations detected by ^1^H NMR-based metabolomics in amino acid and protein metabolism and in ketogenesis. Importantly, it allowed us to discriminate the metabolic abnormalities associated with fatal COVID-19 and to complement the NMR data, providing insightful information regarding the alterations in one-carbon metabolism and in metabolites associated with insulin sensitivity and inflammation. The metabolites related to one-carbon metabolism, glycerophosphocholine, serine, betaine, and histidine, were lower (Figure 3A–D), whereas xanthine and hypoxanthine were higher in infected subjects if compared to controls (Figure 3E,F). The lower levels of glycerophosphocholine in the survivors and non-survivors groups, when compared to control subjects, support the ^1^H NMR-based metabolomics results, which detected lower (CH_3_)_3_ choline-related metabolites in the two groups of infected patients. Metabolites associated with inflammation were significantly higher in infected subjects compared to controls. Importantly, creatinine, 4-hydroxyproline, gluconic acid, and *N*-acetylserine in admission were predictive of ICU-related mortality, being significantly higher in the non-survivors compared to both survivors and control groups (Figure 3G–J). The higher levels of creatinine in non-survivors confirmed the results of the NMR analysis (Figure 2C). Moreover, asymmetric dimethylarginine and methylmalonic acid were significantly higher in survivors and non-survivors when compared to control subjects (Figure 3K,L). MS-based metabolomics revealed significantly higher content of *β*-hydroxybutyrate—a ketone body (Figure 3M) and lower content of the amino acid tryptophan (Figure 3N) in infected subjects if compared to controls. Appendix A shows the essential stability of the QC samples throughout the run, which was within the SD limit. Additionally, the IS p-fluoro-DL-phenylalanine and U-^13^C D-glucose, which were added to all samples, showed that the CV was less than 10%, indicating that the differences observed in metabolites among groups could be attributed to biological variation and not to the variability of the analytical system. Appendix A show the parameters of the assigned metabolites with significant differences among groups according to the non-targeted MS-metabolomics in the negative and positive modes, respectively.

CART models were built to assess the predictive power of metabolite levels in classifying study participants into control, survivors, and non-survivors. These models were built using the data from the NMR and MS metabolomics and considered only the metabolites that showed discriminatory power among the groups (metabolites shown in Figure 2 and Figure 3). Additionally, the subjects’ sex and age were included in the models. Appendix A show the CART models fitted to data indicating that choline-related metabolites at ICU admission had the highest predictive power with a first partition point at ≥0.8813 (normalized data). The classification trees show that over 90% of control subjects followed the classification of ≥0.8813, and most survivors (95%) and non-survivors (89%) were classified as <0.8813. A second partition criteria was observed at creatine/creatinine < 0.1156 (Appendix A) and at *N*-acetyl serine < 0.00137 (Appendix A). For this second partition, over 70% of survivors were included. Conversely, for the non-survivors, the majority (64%) were classified at ≥0.116 for creatine/creatinine and ≥0.00137 for N-acetyl serine. Considering the models using the MS variables, hypoxanthine had the highest predictive power with a first partition point at <−0.2694, where 100% of control subjects followed this classification and over 80% of survivors and non-survivors were classified as ≥−0.2694 (Appendix A). A second partition criteria was observed at *N*-acetylserine < 0.2264, where most survivors were included, whereas over 50% of non-survivors were classified at ≥0.2264. Lastly, a CART model was built with creatinine, 4-hydroxyproline, gluconic acid, and *N*-acetylserine, metabolites that were higher in the non-survivors compared to survivors and controls (Appendix A). This model shows that *N*-acetylserine had the sole predictive power with a first partition point of <0.2264 (100% of control and 94% of survivors). Most non-survivors (65%) were classified at ≥0.2264, confirming the results of the models presented in Appendix A. The second partition point included the majority of control subjects at *N*-acetylserine < −0.5714 and survivors at ≥−0.5714. 

Appendix A present the final allocation contingent among the morbidity classes by applying the classification criteria described above. These results clearly suggest that choline-related metabolites at ICU admission have a protective effect on 28-day outcomes. As opposed to *N*-acetylserine, creatine/creatinine, and hypoxanthine, where higher contents at admission predicted fatal outcomes. 

### 3.3. Lipoprotein Dynamics

A complete characterization of lipoproteins was performed (Figure 4), and significant changes were observed as a function of severe COVID-19. The results show that survivors and non-survivors presented lower concentrations of total cholesterol, LDL, and HDL if compared to controls (Figure 4A–C). VLDL and triacylglycerol concentrations were significantly higher (Figure 4D,E), and non-HDL cholesterol (Figure 4E) was significantly lower in the non-survivors if compared to controls. 

### 3.4. Sex-Based Differences in Lipoproteins and Metabolites

Lastly, we investigated whether changes in lipoproteins and in metabolites associated with severe COVID-19 differed according to sex by two-factor analysis (Figure 5). In general, the changes observed in female subjects mirrored the changes described above when considering all subjects. Our findings indicate that the strongest effect was seen in severe COVID-19 cases, and no differences between men and women within the group were observed. However, for HDL, total cholesterol, non-HDL cholesterol, LDL to HDL, and HDL to cholesterol ratios, significant interactions between disease and sex were observed. Additionally, lower contents of HDL (Figure 5C) and higher contents of VLDL (Figure 5D) and triacylglycerols (Figure 5E) were observed in female but not male subjects in the non-survivors group when compared to controls.

The same sex-related pattern changes were observed for selected metabolites analyzed by ^1^H NMR-based metabolomics, where the strongest effect was seen in severe COVID-19 cases (Figure 5I–L). Lower content of (CH_3_)_3_ choline-related metabolites (Figure 5I) and higher content of *N*-acetyl glycoproteins (Figure 5K) were observed among females in both survivors and non-survivors if compared to controls, whereas among men these differences were observed only between survivors and controls. Additionally, higher contents of acetoacetate (Figure 5J) and creatine/creatinine (Figure 5L), as a function of severe COVID-19, were only observed in women but not in men. Indeed, for acetoacetate and creatine/creatinine, significant interactions between disease and sex were observed. There were no significant differences in amino acids and one-carbon metabolism-related compounds when sex was considered a variable. The metabolic alterations observed in severe COVID-19 are shown in Figure 6.

## 4. Discussion

The present study aimed to investigate plasma metabolic changes in severe COVID-19 patients at admission that are associated with ICU-related mortality. Our goal was to search for potential metabolic pathways that could be involved in severe COVID-19 pathophysiology and disease outcome. We identified significant changes in a plethora of metabolites, indicating that severe COVID-19 dysregulates one-carbon, lipid, and amino acid metabolism and lipoprotein dynamics.

Higher contents of creatine/creatinine, 4-hydroxyproline, gluconic acid, and *N*-acetylserine in non-survivors if compared with survivors and control groups indicate that these metabolites are associated with uncontrolled inflammation, multi-organ dysfunction, particularly liver and kidneys, and some degree of insulin resistance associated with fatal COVID-19 outcomes. They may be considered biomarkers for prognostic purposes and to monitor how different organs and tissues respond to severe infection. For instance, gluconic acid has been previously linked to hyperglycemia and brain injury in ischemic stroke [53], and it may be considered a marker of oxidative stress. Additionally, *N*-acetylation of amino acids, including the formation of *N*-acetylserine, has been associated with SARS-CoV-2 infection and COVID-19 pathogenesis [54], whereas higher levels of *N*-acetylserine have been recently considered a marker of the progression of chronic kidney disease [55]. Our data, built on these previous observations, highlights an association between these metabolic alterations and poor outcomes in a cohort of ICU-admitted patients.

Higher plasma contents of creatine/creatinine can be an indication of lower sensitivity to insulin, as we observed in the fatal cases of COVID-19. Although higher creatine levels in severe COVID-19 have been associated with kidney dysfunction [26] and are important for viral replication [56], recent findings place creatine as a key metabolite involved in the regulation of adipocyte thermogenesis, whole-body energy metabolism, and immunity [57,58]. Therefore, higher levels of creatine/creatinine in infected subjects can be regarded as a biomarker of detrimental effects on metabolic health and immune responses imposed by SARS-CoV-2 infection.

Higher levels of 4-hydroxyproline found in non-survivors are a strong indication of disturbed amino acid metabolism induced by SARS-CoV-2 infection, as already suggested [20,23,24,25]. In humans, 4-hydroxyproline in the blood is a product of protein degradation, mainly collagen. Most 4-hydroxyproline is recycled back to the liver and kidneys to synthesize glycine, and this seems to be an important source of glycine for cells and tissues [59]. Therefore, the significantly higher levels of 4-hydroxyproline in non-survivors can be regarded as a feature of severe COVID-19, suggesting liver dysfunction and a lower availability of glycine. This result reveals a potential disruption in the one-carbon metabolism pathway, which is significant as glycine serves as a crucial methyl donor in this pathway. This disruption has already been described previously in cells infected with SARS-CoV-2 in vitro [27] and confirmed in patients with severe COVID-19 in the present study and mild to moderate COVID-19 in another [26]. 

In addition to glycine, significant lower contents of serine, betaine, and histidine, which are metabolites also involved in one-carbon metabolism, were found in infected subjects. Importantly, the significant drop in choline-related metabolites (as observed in the NMR-based metabolomics), which agrees with a drop in glycerophosphocholine (by MS-based metabolomics), strengthens the idea that severe COVID-19 alters one-carbon metabolism and affects the availability of methyl donors. Indeed, a comprehensive characterization of the alterations in one-carbon metabolism would help integrate the metabolic changes in glucose, amino acid, nucleotide, and lipid metabolism [60], as well as provide a better understanding of the alterations in epigenetic regulation and redox homeostasis associated with severe COVID-19. 

Phosphatidylcholine is the most abundant phospholipid, and in humans, its synthesis is probably the main point of deviation for one-carbon donors [61]. Our results show that severe COVID-19 alters lipoprotein dynamics, as indicated by lower contents of total cholesterol and HDL- and LDL-cholesterol and higher contents of VLDL and triacylglycerols, particularly in fatal COVID-19 (Figure 4). Therefore, one may speculate that the alterations in one-carbon metabolism observed in infected subjects caused a drop in choline-related metabolites, which significantly disrupted lipoprotein dynamics, as indicated by other studies [61,62]. Conversely, lower choline availability may have contributed to reduced phospholipid synthesis by the liver and disturbed lipoprotein dynamics, as choline deficiency is known to alter lipid metabolism and induce liver fibrosis [62]. Another indication of the effect of SARS-CoV-2 infection on host lipid metabolism and lipoprotein dynamics was recently shown by our group, where simvastatin, by disrupting lipid rafts in human epithelial lung cells, prevented SARS-CoV-2 entry and replication [63]. Importantly, NMR-based metabolomics has also shown that alterations in lipoprotein dynamics may be associated with the systemic effects of COVID-19, even in subjects in the recovery phase of the disease. Particularly, the HDL subfraction negatively correlates with inflammatory cytokines [64]. Additionally, apolipoprotein alterations are more pronounced in the fatal cases of COVID-19 [65]. 

This scenario is compatible with the CART models fitted to data where choline-related metabolites had a protective effect at ICU admission (Appendix A). Recent evidence links choline metabolism in the liver and the gut microbiota to endothelial function and thrombosis [66,67], which is one of the clinical manifestations associated with disease progression and severe COVID-19 [68]. Indeed, choline seems to be essential to sustain mitochondrial energetics and maximal platelet activation, and, therefore, to regulate thrombosis [69]. Additionally, CART models indicated that higher levels of *N*-acetylserine, creatine/creatinine, and hypoxanthine at admission are strong predictors of COVID-19 fatality (Appendix A). These results add to the discussion about the detrimental effects of severe COVID-19 on liver and kidney function and metabolic health.

We also found increased levels of asymmetric dimethylarginine in infected subjects, which is produced by arginine methylation. Therefore, asymmetric dimethylarginine synthesis is directly related to one-carbon metabolism for methyl donor availability. Additionally, asymmetric dimethylarginine has been strongly associated with fibrosis in the liver, kidney, and heart [70,71], tissues that are compromised in COVID-19. Therefore, these data strongly support that alterations in one-carbon and amino acid metabolism are linked to liver damage in the most severe outcomes of COVID-19.

Our study confirmed that severe COVID-19 induced important changes in amino acid metabolism, where significantly lower contents of alanine, BCAA (valine, isoleucine, and leucine), glutamine, tryptophan, glycine, and tyrosine were found in both survivors and non-survivors if compared to control subjects. Lower amino acid levels may indicate an increase in amino acid catabolism to provide the necessary supply of carbon and ATP to support viral replication [16,72,73]. For instance, isoleucine [74] and glutamine [75] seem to be essential for SARS-CoV-2 replication. In the case of glutamine, it has been shown that inhibition of glutaminolysis halted SARS-CoV-2 in primary astrocytes in a rodent model; the authors suggested that lower availability of glutamine is a contributing factor for the neurological impairments observed in long post-acute COVID-19. As observed in the plasma of infected subjects, lower contents of salivary amino acids are associated with SARS-CoV-2 infection [76]. Additionally, a decrease in plasma glutamine [77] and lower salivary levels of tyrosine and BCAA [78] seem to be associated with COVID-19 severity, which is consistent with our findings. Despite the fact that the literature shows an association between levels of circulating amino acids and COVID-19 severity, amino acid levels seem not to be good predictors of ICU admission or disease fatality [79]. 

Lower levels of glutamine in the plasma may indicate increased use of this amino acid, especially by the host’s liver and immune cells. This increased use may be needed to support the synthesis of proteins and inflammatory mediators during the acute phase of infection. Indeed, the liver is actively involved in the synthesis of acute-phase proteins. In this context, the assigned broad signals, ƍ^1^H = 2.04–2.08 ppm, of *N*-Acetyl glycoproteins, which were higher in severe COVID-19 subjects (Figure 2D), may include signals from the acute-phase proteins, such as α1-acid glycoprotein, α1-antitrypsin, and haptoglobin, and to the ^1^H from sidechains of *N*-acetyl-glucosamine and *N*-acetylneuraminic acid [80]. ^1^H signals of *N*-acetyl glycoproteins have been suggested to be markers of SARS-CoV-2 infection and inflammation and to be implicated in long post-COVID symptoms as well [81]. Indeed, enrichment analysis of genes associated with cases of severe COVID-19 indicates acute phase response and inflammation among the most significant biological functions altered, according to a recent multi-omics analysis of COVID-19 datasets [82]. 

Other authors have shown that liver infection by SARS-CoV-2 directly contributes to hepatic dysfunction and that deceased subjects often presented abnormal liver enzymes, microvesicular steatosis, and mild inflammatory infiltrates in the hepatic lobule and portal tract [83,84]. The increase in purine metabolites (xanthine and hypoxanthine) may also be regarded as a direct effect of SARS-CoV-2 infection in the liver that enables virus replication. Higher levels of deoxycytidine, associated with increased viral load, have already been associated with severe and fatal outcomes [85]. Accordingly, the liver of patients with severe COVID-19 shares many similarities with that of non-alcoholic fatty liver disease (NAFLD), a common manifestation of the metabolic syndrome that can progress to hepatocyte injury, inflammation, and fibrosis [86,87].

Lastly, the higher levels of acetoacetate and ß-hydroxybutyric acid observed in infected subjects also reflect the impact of severe COVID-19 on liver function. Dysregulation in ketogenesis is also associated with the pathogenesis of NAFLD and decreased insulin sensitivity, an important manifestation of the metabolic syndrome. In this scenario, enhanced ketogenesis seems to be a consequence of increased influx and accumulation of lipids in the liver (e.g., triacylglycerols), which in turn results in an increased flux of Acetyl-CoA [62]. Metabolites’ profiles in the urine of COVID-19 subjects are also compatible with enhanced ketogenesis, as shown by higher excretion of carnitine and acetone in the acute phase of the disease compared to the recovery period and control subjects [88]. In addition, our group has recently pointed out the central role of the acetyl-CoA pathway in the immunometabolism response associated with the Sinovac vaccine [89].

The metabolic disturbances observed in COVID-19 subjects seem to be influenced by the SARS-CoV-2 variant and clinical presentation [90], which highlights the novelty of the results presented in this study. Interestingly, there have been some suggestions that the metabolic disturbances observed in hospitalized subjects with COVID-19 are dependent on the wave of the infection [91]. In this regard, arginine and threonine were altered in the early wave (between May and July 2020) but not in the latter wave (September 2020 to June 2021) of COVID-19 in that particular study. On the other hand, a machine learning approach identified the same pattern of change in glutamate, aspartate, glycolithocholic acid, and methionine sulfoxide across both waves of COVID-19, corroborating our results.

Our results show that alterations in key metabolites, such as choline metabolites, creatinine/creatinine, ^1^H signals of *N*-acetyl of glycoproteins, and acetoacetate, as well as changes in lipoproteins, are greater in women. Current evidence indicates that men are more vulnerable to severe COVID-19, and higher mortality rates have been observed in this category [92]. Male subjects presented significant alterations in the tryptophan-kynurenine pathway and plasmalogen, which were associated with increased inflammation and stress biomarkers [93,94]. On the other hand, women seem to be more susceptible to developing post-acute COVID syndrome [95]. Therefore, our study highlights the importance of considering the sex-based differences here described when monitoring and treating COVID-19 patients.

## 5. Conclusions

This is the first study to demonstrate host metabolic disturbances associated with severe COVID-19 and to investigate the responses in the fatal cases of the disease in hospitalized subjects in Rio de Janeiro during the early months of the COVID-19 pandemic in Brazil. While our study provides new insights into the metabolic disturbances associated with fatal outcomes of COVID-19, one limitation is its reduced sample size. We identified that severe COVID-19 dysregulates one-carbon, lipid, and amino acid metabolism and lipoprotein dynamics. The higher contents of creatine/creatinine, 4-hydroxyproline, gluconic acid, and *N*-acetylserine observed in the fatal COVID-19 outcome may be associated with uncontrolled inflammation, multi-organ dysfunction, and some degree of insulin resistance. Since the incidence of severe outcomes after hospital discharge can be very high in Brazil [28], these metabolic alterations may be considered to improve our understanding of the pathophysiology of post-acute COVID-19 syndrome, as already suggested for the metabolic alterations associated with post-acute cardiovascular events [96]. Additionally, the sex differences observed in our study should also be considered when designing strategies for pandemic surveillance. Doing so may lead to better disease stratification and improved patient outcomes.

## Figures and Tables

**Figure 1 metabolites-13-00879-f001:**
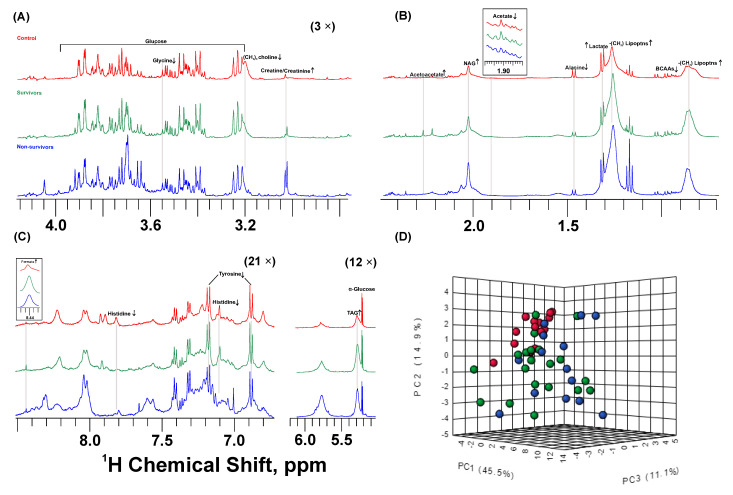
^1^H NMR-based metabolomics shows different plasma metabolite profiling in severe COVID-19 patients compared to control subjects. ^1^H NMR representative spectra of control (red), COVID-19 survivors (green), and non-survivors (blue). Metabolites that differ significantly among groups are indicated as having higher (↑) or lower (↓) contents compared to controls; (**A**,**B**) aliphatic region (3× magnified); (**C**) amidic and aromatic regions (21× magnified); (**D**) Principal Component Analysis 3D score plot shows discriminating profiling among groups; control (red), COVID-19 survivors (green), and non-survivors (blue).

**Figure 2 metabolites-13-00879-f002:**
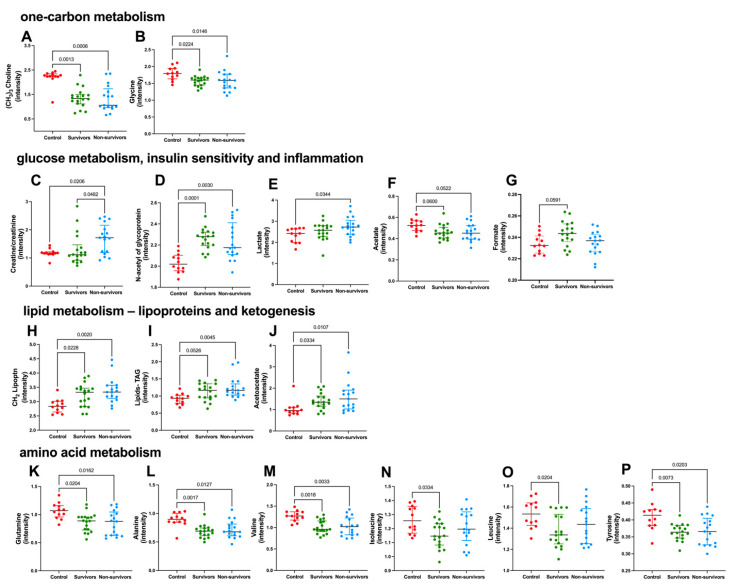
Metabolites that were significantly altered in severe COVID-19, according to ^1^H NMR-based metabolomics. Most discriminating metabolites according to PCA loading factors, presenting significant differences among groups. Control, n = 12 (red circles); survivors, n = 18 (green circles); non-survivors, n = 17 (blue circles). Metabolites related to one-carbon metabolism (**A**,**B**): (CH_3_)_3_ choline-related metabolites and glycine; metabolites related to glucose metabolism, insulin sensitivity, and inflammation (**C**–**G**): creatine/creatinine, *N*-Acetyl of glycoproteins, lactate, acetate, and formate; metabolites related to lipid metabolism (**H**–**J**): CH_2_ lipoproteins (mainly VLDL), lipids-TAG (CH=CH olefinic protons of triacylglycerols) and acetoacetate; metabolites related to amino acids and protein metabolism (**K**–**P**): glutamine, alanine, valine, isoleucine, leucine, and tyrosine; metabolites’ contents were determined according to their respective peak intensity. Data were presented as medians with an interquartile range, and only significant *P* values are shown, according to Kruskal-Wallis and Dunn’s post-hoc tests.

**Figure 3 metabolites-13-00879-f003:**
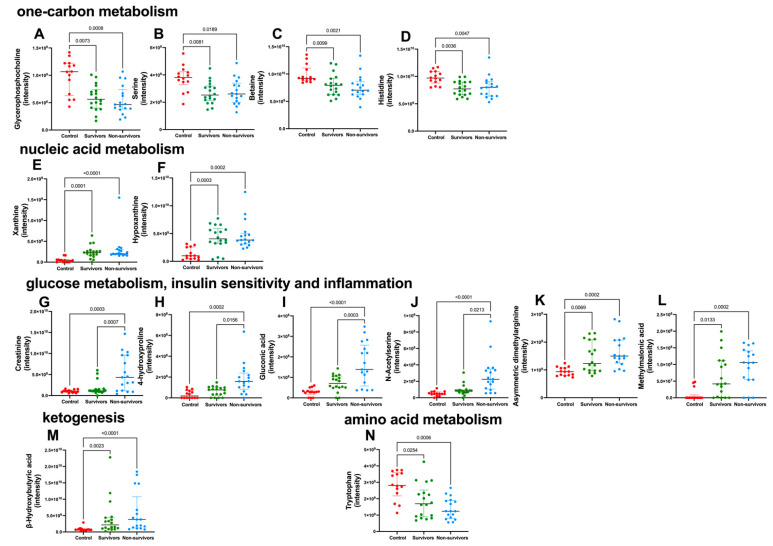
High-resolution mass spectrometry-based metabolomics shows an altered plasma metabolic profile in severe COVID-19. Assigned metabolites in the untargeted approach and confirmed in the targeted approach, presenting significant differences among groups. Control, n = 12 (red circles); survivors, n = 18 (green circles); non-survivors, n = 17 (blue circles). Metabolites related to one-carbon metabolism (**A**–**D**): glycerophosphocholine, serine, betaine, histidine, and nucleic acid metabolism (**E**,**F**): xanthine, hypoxanthine; metabolites related to glucose metabolism, insulin sensitivity, and inflammation (**G**–**L**): creatinine, 4-hydroxyproline, asymmetric dimethylarginine, gluconic acid, *N*-acetyl serine, and methylmalonic acid. Metabolite related to lipid metabolism (**M**): *β*-hydroxybutyrate; metabolite related to amino acid metabolism (**N**): tryptophan. Metabolites’ contents were determined according to their respective peak intensities. Data were presented as medians with an interquartile range, and only significant *p* values are shown, according to Kruskal-Wallis and Dunn’s post-hoc tests.

**Figure 4 metabolites-13-00879-f004:**
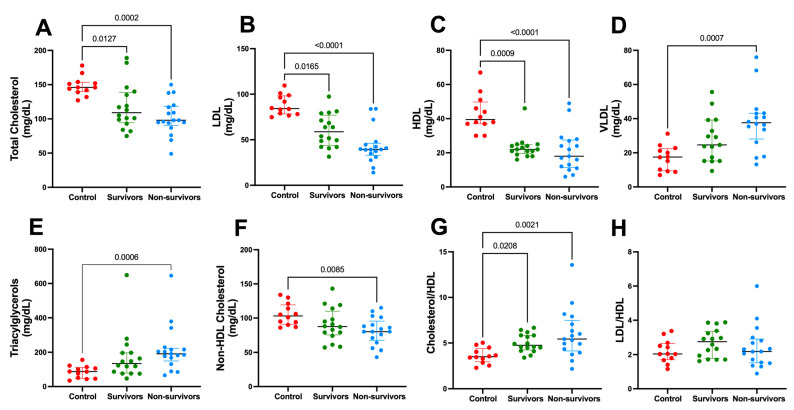
Lipoprotein dynamics changed significantly in severe COVID-19. (**A**) Total Cholesterol; (**B**) Low-Density Lipoprotein—LDL; (**C**) High-Density Lipoprotein—HDL; (**D**) Very Low-Density Protein—VLDL; (**E**) Triacylglycerol; (**F**) Non-HDL Cholesterol; (**G**) Cholesterol-to-HDL ratio; and (**H**) LDL-to-HDL ratio. Control, n = 12 (red circle); survivors, n = 18 (green circle); non-survivors, n = 17 (blue circle). Data were presented as medians with an interquartile range, and only significant *p* values are shown, according to Kruskal-Wallis and Dunn’s post-hoc tests.

**Figure 5 metabolites-13-00879-f005:**
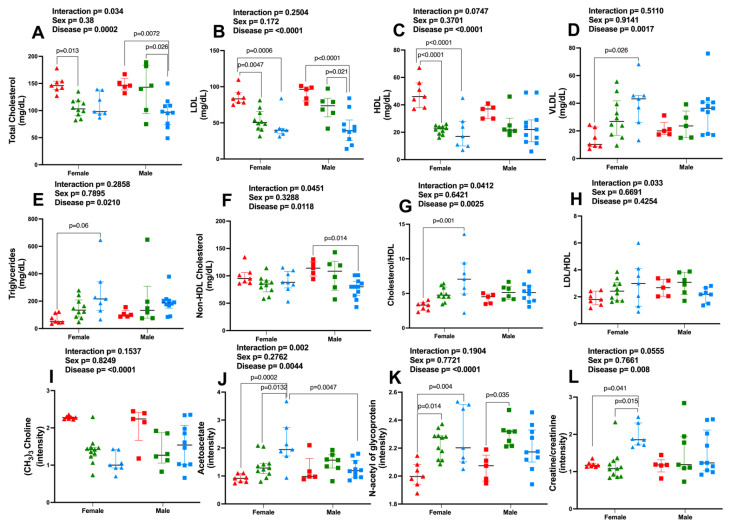
Severe COVID-19-induced changes in lipoproteins and metabolites are greater in women than in men. (**A**) Total Cholesterol; (**B**) Low-Density Lipoprotein—LDL; (**C**) High-Density Lipoprotein—HDL; (**D**) Very Low-Density Protein—VLDL; (**E**) Triacylglycerol; (**F**) Non-HDL Cholesterol; (**G**) Cholesterol-to-HDL ratio; (**H**) LDL-to-HDL ratio; (**I**) (CH_3_)_3_ Choline; (**J**) Acetoacetate; (**K**) *N*-acetyl of glycoprotein; and (**L**) Creatine/creatinine. Female subjects: control, n = 7 (red triangles); survivors, n = 11 (green triangles); non-survivors, n = 7 (blue triangles). Male subjects: control, n = 5 (red squares); survivors, n = 7 (green squares); non-survivors, n = 10 (blue squares). Data were presented as medians with an interquartile range, and *p* values are shown according to a two-factor ANOVA and Tukey’s multiple comparison tests.

**Figure 6 metabolites-13-00879-f006:**
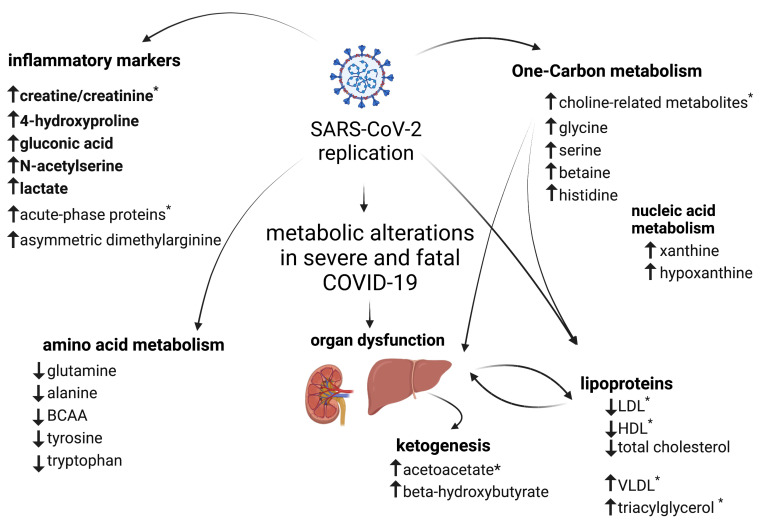
Metabolic alterations in severe COVID-19. Summary of changes in plasma metabolites in survivors and non-survivors. Metabolites that differ significantly among groups are indicated as having higher (↑) or lower (↓) contents compared to controls. Metabolites in bold were higher in the fatal cases of the disease compared to survivors and controls. * indicates metabolites with changes greater in women than in men. This figure was created with BioRender.com.

**Table 1 metabolites-13-00879-t001:** Demographics and clinical characteristics of control, COVID-19 survivors, and non-survivors.

	Control (n = 12)	Survivors (n = 18)	Non-Survivors (n = 17)	*p* Value
Age, years	50 (36–59)	56 (39–63)	58 (51–73)	0.344
Sex, male; n (%)	5 (41)	7 (38)	10 (58)	0.727
Respiratory support; n (%)				
Noninvasive O_2_ supplementation	0 (0)	8 (44)	0 (0)	**0.003**
Mechanical ventilation	0 (0)	10 (66)	17 (100)	
SAPS II	n.a.	55 (37–64)	68 (59–78.5)	**0.001**
PaO_2_/FiO_2_ ratio	n.a.	196 (154–429)	139 (177–178)	0.099
Time from symptom onset to blood sample (days)	n.a.	10 (7–14)	10 (3–14)	0.975
**Comorbidities; n (%)**				
Obesity	1 (8.3)	5 (27.7)	2 (11.7)	0.650
Hypertension	2 (16)	4 (22)	5 (29)	0.855
Diabetes	0 (0)	6 (33)	6 (35)	0.990
Cancer	0 (0)	2 (11)	2 (11)	0.990
Heart disease ^1^	0 (0)	2 (11)	2 (11)	0.990
**Laboratory findings at admission**				
Leukocytes, ×1000/µL	n.a.	12.4 (9.1–14.5)	14.8 (11.5–21.7)	0.097
Lymphocytes, cells/µL	n.a.	1288 (939–1.579)	1035 (284–1.706)	0.521
Monocytes, cells/µL	n.a.	495 (448–742)	738 (599–1.005)	**0.009**
Platelet count, ×1000/µL	n.a.	198 (154–324)	187 (131–240)	0.125

Continuous variables are represented as the median and interquartile range. Categorical variables are represented as *n* (frequency %). n.a.—not applicable ^1^ Coronary artery disease or congestive heart failure. Categorical variables were compared using the two-tailed Fisher exact test, and continuous variables were compared using student’s *t* or ANOVA tests for parametric and Mann-Whitney U or Kruskal-Wallis tests for nonparametric distributions. Significant *p* values are in bold.

## Data Availability

The NMR data presented in this study was assigned using COLMARm, https://spin.ccic.osu.edu/index.php/colmarm, session ID 3121-pZ5ZukwXBh, (accessed on 14 July 2023). The raw NMR and MS data presented in this study are available on request from the corresponding author. The data are not publicly available as multivariate analyses are being performed.

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
