# Peer review of "Integrated NMR and MS Analysis of the Plasma Metabolome Reveals Major Changes in One-Carbon, Lipid, and Amino Acid Metabolism in Severe and Fatal Cases of COVID-19"

_metabolites, 2023, doi:10.3390/metabo13070879_

Round 1

Reviewer 1 Report

In this manuscript, authors try to find the bio markers related to the Covid-19 patient. They focus on the plasma metabolism using NMR and MS analyses. Turn out, the interesting targets are involved in one-carbon, lipid and amino acids metabolic pathways. Overall authors did a good job, the methods and results are clearly expressed. However, the number of statistic population seem to be a little too few. To increase the experimental number, under a pandemic, it should be not very difficulty. Moreover, it is known there are so many Covid-19 variants, the distinct variant causes some different syndrome, so, it might affect the different metabolic pathways. Authors did not mention what Covid-19 variant been used in their study. Authors should address this issue that will improve this manuscript much higher impact.

Reviewer 2 Report

Aside from the fact that the sample size is on the low side, which is probably inherent to this type of studies, the work and the ms are well done.

Very minor remarks:

English language needs some moderate polishing (commas)

What is »severe COVID« (as compared to non-severe COVID)?

»600 L« > 600 µl

»MHZ« > MHz

English language needs some moderate polishing (commas), but is OK in general

Reviewer 3 Report

Thank you for submitting the manuscript entitled “Integrated NMR and MS analysis of plasma metabolome reveals major changes in one-carbon, lipid, and amino acid metabolisms in Severe and Fatal COVID-19 Subjects”

The paper is well written with very interesting results.

I have carefully reviewed your manuscript and would like to provide feedback and suggestions for enhancement. 

My comments are as follows. 

The title should be reconsidered and rephrased. Including the methods of analysis in the title is optional. Moreover, using the term “fatal cases” instead of “fatal subjects” would be more appropriate. 

Line 38, 40: it is advisable for the authors to present the complete words when they are mentioned for the first time. Additionally, it is essential to define these abbreviations in parentheses the first time they appear in the abstract, main text, and figure or table captions.

Line 46: it is crucial for the authors to provide more detailed information regarding the statement that the incidence of severe outcomes after hospital discharge is very high in Brazil. Similarly, additional elaboration is necessary to support the claim that Brazil has one of the highest mortality rates. These statements require supporting evidence, such as specific data, references, or relevant studies, to comprehensively understand the situation. 

Line 113: Instead of using the phrase “in this study we…” it would be more effective for the authors to articulate the purpose or objective of their study clearly. In this way, they can provide a concise and focused introduction that guides readers and helps them understand the research's specific goals, enhancing the clarity and coherence of the manuscript. 

Line 137: A definition for severe cases and a reference is needed.

Line 146: The authors need to provide more specific details regarding the treatment protocol followed in the study and the rationale behind the decision not to administer antivirals, steroids, or other anti-inflammatory or antiplatelet drugs to the patients. Additionally, it would be valuable for the authors to clarify whether their treatment practice was aligned with the international guidelines that were in place at the time of the study. 

Line 306: Instead of stating “a total of 47 plasma samples”, it would be more appropriate to say “a total of 47 subjects” included in the study. 

In the conclusions, it is valuable to present and discuss the specific results of this study. It is more comprehensive for the readers to compare, agree, and disagree with the existing literature presented in the discussion section. 

Minor editing of the English language required

Reviewer 4 Report

In the manuscript Integrated NMR and MS analysis of plasma metabolome reveals major changes in one-carbon, lipid, and amino acid metabolism in severe and fatal COVID-19 subjects, the authors describe detailed metabolic analysis of severe COVID patients blood sera. The implementation of complimentary analytical techniques like NMR and liquid chromatography, provide insight into the possible mechanisms linked with the severe COVID. The study also provides a basis for the future identification of metabolic biomarkers related to the outcomes of severe COVID. Although the study presents a welcome addition to the understanding of COVID, some clarifications are necessary to improve the quality of the paper.

1.      I was not able to access the list of the assigned metabolites in COLMAR using the provided code. Please, check the code or provide the list of metabolites in alternative manner.

2.      Was any normalization and/or scaling applied to data prior to statistical analysis?

3.      The authors also performed the analysis of lipoproteins, however, no details were provided in the Methods.  

4.      In building the CART model, the authors chose to include only the metabolites identified as significant by NMR analysis. Considering the complementarity of used analytical methods, why were the differentially expressed metabolites identified by LCMS as well as lipoproteins not included in the model? Also, only univariate analysis seems to be applied to LCMS data.

5.      Although the Conclusions emphasize the significance of differential metabolic response to severe COVID between men and women, the Discussion hardly touches on the issue, and it is not mentioned at all in the Abstract.

6.      Additionally, throughout the text the authors refer to the long post-COVID. However, it is not clear at all how the results of the present study are related to the understanding of long post-covid pathophysiology due to the lack of patient follow-up.

Minor:

Please, remove lines 123-124, 297-303, and 317-319 from the text. 

Round 2

Reviewer 1 Report

In this revised version manuscript, authors do some changes to response the reviewers’ comments. It is good, however, authors make a “new” title that is different from the title of other related submitted revised documents! I am just curious about whether the “new” title of revised version is really real new one or just one of the “older” version manuscripts and been suddenly showed again. Anyway, authors have to carefully to make all titles in every submitted documents are same. Another comment, since author only described “The most prevalent SARS-CoV-2 variant circulating in Brazil during the period this cohort took place was the gamma variant”, they did not mention whether or not to identify the genomic type of samples at that time (I guess not). As if really not, I strongly suggest authors should honestly to tell readers they did not do the COVID-19 variant identification of collected samples. Probably due to they did not except the COVID-19 variants are so easily generated and cause so diversity of syndrome during that period of time.
